# Glucose Transporter 9 (GLUT9) Plays an Important Role in the Placental Uric Acid Transport System

**DOI:** 10.3390/cells11040633

**Published:** 2022-02-11

**Authors:** Benjamin P. Lüscher, Christiane Albrecht, Bruno Stieger, Daniel V. Surbek, Marc U. Baumann

**Affiliations:** 1Department of Obstetrics and Gynecology, University Hospital of Bern, University of Bern, CH-3010 Bern, Switzerland; l_beni@hotmail.com (B.P.L.); daniel.surbek@insel.ch (D.V.S.); 2Department of Biomedical Research, University Hospital of Bern, University of Bern, CH-3010 Bern, Switzerland; 3Institute of Biochemistry and Molecular Medicine, University of Bern, CH-3012 Bern, Switzerland; christiane.albrecht@ibmm.unibe.ch; 4Swiss National Center of Competence in Research, NCCR TransCure, University of Bern, CH-3012 Bern, Switzerland; bruno.stieger@uzh.ch; 5Department of Clinical Pharmacology and Toxicology, University Hospital of Zürich, University of Zurich, CH-8091 Zurich, Switzerland

**Keywords:** glucose transporter 9, GLUT9, uric acid, preeclampsia

## Abstract

Background: Hyperuricemia is a common laboratory finding in pregnant women compromised by preeclampsia. A growing body of evidence suggests that uric acid is involved in the pathogenesis of preeclampsia. Glucose transporter 9 (GLUT9) is a high-capacity uric acid transporter. The aim of this study was to investigate the placental uric acid transport system, and to identify the (sub-) cellular localization of GLUT9. Methods: Specific antibodies against GLUT9a and GLUT9b isoforms were raised, and human villous (placental) tissue was immunohistochemically stained. A systemic GLUT9 knockout (G9KO) mouse model was used to assess the placental uric acid transport capacity by measurements of uric acid serum levels in the fetal and maternal circulation. Results: GLUT9a and GLUT9b co-localized with the villous (apical) membrane, but not with the basal membrane, of the syncytiotrophoblast. Fetal and maternal uric acid serum levels were closely correlated. G9KO fetuses showed substantially higher uric acid serum concentrations than their mothers. Conclusions: These findings demonstrate that the placenta efficiently maintains uric acid homeostasis, and that GLUT9 plays a key role in the placental uric acid transport system, at least in this murine model. Further studies investigating the role of the placental uric acid transport system in preeclampsia are eagerly needed.

## 1. Introduction

Preeclampsia, characterized by hypertension and proteinuria during pregnancy, contributes substantially to perinatal morbidity of both the mother and her child. There is a growing body of evidence that indicates that uric acid plays a role in the pathogenesis of preeclampsia. [1,2] Uric acid is the final metabolic product of purine metabolism in humans and great apes. The plasma uric acid concentration in humans is higher (180–420 µmol/L) than in other mammalian species (30–120 µmol/L) [3], which is due to the mutational silencing of the liver enzyme uricase. Uric acid levels are regulated within relatively tight limits. Although it is a potent antioxidant, its limited solubility leads to metabolic disease such as diabetes, gout, kidney stone disease and hypertension, already with small increases above normal levels [4,5,6,7,8,9]. Plasma urate levels decrease in the first trimester of pregnancy by at least 25%, return to normal levels in the second trimester and then increase towards the end of pregnancy [10]. In pregnancies complicated by preeclampsia, elevated uric acid serum levels are commonly observed [1]. An elevated uric acid serum concentration during early pregnancy is considered to be an independent risk factor of gestational hypertension and preeclampsia [11]. Of note, several studies have shown a correlation between the severity of the disease, fetal outcome and uric acid serum levels [12,13,14,15].

The exact mechanisms of the transplacental uric acid transport system remain unclear. Glucose transporter 9 (GLUT9, also known as SLC2A9) is a member of the glucose transporter family. Its sequence is similar to other members of the GLUT family. However, the glucose transport activity of GLUT9 is low, although GLUT9 was shown to be a high-capacity urate transporter [16]. There are two splice variants, GLUT9a and GLUT9b, which differ in the N-terminal domain [17,18]. GLUT9a consists of 12 exons with a length of 540 amino acids, while GLUT9b consists of 13 exons with a total length of 512 amino acids.

The aim of our study was to investigate the placental uric acid transport system using immunohistochemical staining to localize GLUT9a and GLUT9b, and to analyze the transplacental transport capacity in vivo using a murine knockout model. Hence, we used a systemic GLUT9 knockout (G9KO) mouse model. We hypothesized that homozygous knockout fetuses lacking GLUT9 in the liver, as well as in their placentae, would develop hyperuricemia.

## 2. Materials and Methods

### 2.1. Placental Tissue

Placentae from normal pregnancies were collected following elective caesarean sections. The exclusion criteria were fetal anomalies, intrauterine growth restriction, diabetes, (pregnancy-induced) hypertension, anemia, infectious disease, drug use or other medical or obstetric complications. Written informed consent was obtained using a protocol approved by the ethical committee of the Canton of Berne, CH-3010 Bern, Switzerland IRB#178/03, approved on 18 November 2013.

### 2.2. Antibodies

For generating polyclonal antibodies against GLUT9a and GLUT9b, peptides coupled C-terminally to keyhole limpet hemocyanin (KLH) were obtained from PolyPeptide Laboratoire France, Strasbourg, France. The following peptides were used: GLUT9a: MARKQNRNSKELGLVC; GLUT9b: MKLSKKDRGEDEESDC. Rabbit antisera were generated using a protocol described previously [19]. The animal experiments were approved by the local authority supervising animal studies.

### 2.3. Immunofluorescence

Human embryonic kidney cells (HEK-293) were cultured in minimal essential medium (Invitrogen) containing 10% heat-inactivated fetal bovine serum albumin (FBS), 50 units/mL penicillin and 50 μL/mL streptomycin. The cells were plated on 30 mm dishes and transfected with a total amount of 1 μg of DNA/dish and 3 μL lipofectamin 2000/dish (Invitrogen). The cells were fixed on glass cover slips in ice-cold methanol (−20 °C) for 20 s. Following washing with phosphate-buffered saline (PBS) (Inselspital, CH-3010 Bern, Switzerland), the cells were incubated in the presence of the primary antibodies, i.e., rabbit anti-hGLUT9a or −9b antibodies (1:500 diluted in PBS + 0.5% Bovine serum albumin (BSA; Sigma, Switzerland)), for 30 min at 4 °C. After washing three times with ice-cold PBS + 0.5% (*w*/*v*) BSA (PBS/BSA), the cells were incubated with the secondary antibody Alexa fluor 488 goat anti-rabbit IgG (H + L) A11008 (Cross-Adsorbed Secondary Antibody, Alexa Fluor 488 from Thermo Fisher Scientific, catalog # A-11008, RRID AB_143165 Invitrogen, Life Technologies Europe B.V., CH-6300 Zug, Switzerland), for 40 min at 4 °C (1:300 in PBS/BSA). After washing three times with ice-cold PBS/BSA, the slides were analyzed using a LEICA DM6000B microscope.

### 2.4. Immunohistochemistry

The placentae were cut and washed in cold PBS. Villous tissue was dissected into 5–10 mm fragments for histological examination. Villous tissue was fixed in a formaldehyde solution (4% (*v*/*v*); Merck, Whitehouse Station, NJ, USA) for 2–4 h at room temperature (RT), followed by 4 °C for a total time of 24–48 h. Fixed placentae were embedded in paraffin and sectioned into 3 µm coronal slices. After deparaffinization of the slides, the target was retrieved in a Tris-EDTA buffer (10 mM Tris Base, 1 mM EDTA solution, 0.05% (*w*/*v*) Tween 20, pH 9.0) via heat treatment in a pressure cooker for 15 min. The slides were washed in PBS and Tween 20 0.1% and blocked with goat serum 10% (*v*/*v*) and BSA 1% (*w*/*v*) in PBS. Human GLUT9 isoforms were detected with rabbit anti-hGLUT9a and −9b antisera (see above) and the Dako Cytomation EnVision System-HRP (DAKO, Glostrup, Denmark). The slides were washed in PBS and Tween 20 0.1% (2 × 5 min) and incubated with the endogenous peroxidase block solution for 15 min at room temperature. A peroxidase-labeled polymer was applied to the slides for 30 min at RT, followed by 3 washes in PBS (5 min each) and the addition of 3,3′-diaminobenzidine in chromogen solution in buffer substrate for 10–30 min, according to the manufacturer’s instructions. Slides were rinsed in H_2_O, counterstained with hematoxylin (Sigma-Aldrich, St. Louis, MO, USA) for 2 min, then rinsed with tap water for 1 min, dehydrated in a series of ethanol baths (70%, 95%, 100%, *v*/*v*) and xylene and mounted with Eukitt (Sigma-Aldrich, St. Louis, MO, USA). A negative control performed in the absence of the primary antibody revealed no detectable signal (Appendix A).

### 2.5. G9KO Mice

G9KO mice were a generous gift from Bernard Thorens (Center for Integrative Genomics, University of Lausanne, Lausanne, Switzerland) and have been characterized previously [20]. Mice were kept in colonies of 5 mice per cage with 12 h day/night cycles and free access to food and water. Female mice were prepared for mating by placing the animals in a cage where male animals had been kept previously, 3 days ahead of mating. For mating, 2 female animals were placed together with 1 male animal. On the next day at 7:30 AM, the animals were separated, and females were checked for plaque formation. Plaque detection was defined as being on day 0.5 of gestation. The mating of heterozygous GLUT9+/− animals resulted in a typical Mendelian off-spring ratio of 25:50:25 for (wild type):(heterozygous):(knockout).

Heterozygous animals were crossbred to obtain wild-type (WT), which also served as controls, and knockout (KO) pups within the same litter. Maternal mice were fed with either standard chow or standard chow +1 g/kg inosine after mating. At day 18.5 after mating, maternal mice were sacrificed, and uric acid serum levels were measured. Fetuses from these animals were isolated and blood was collected via decapitation for uric acid measurements. Plasma uric acid was analyzed using a Roche/Hitachi 902 robot system (Roche); no dilution of fetal sera was needed.

### 2.6. Statistical Analysis

The investigators performing the statistical analysis were blinded. Data are expressed as the mean ± SEM. Graph prism software was used for statistical analysis. Differences between means were tested using Student’s *t*-test or by linear regression analysis where appropriate. A *p*-value of <0.05 was considered to be significant.

## 3. Results

### 3.1. Localization of GLUT9 Isoforms in Human Placenta

Villous (placental) tissue was embedded in paraffin and histological sections were prepared. Immunohistochemical staining against GLUT9b revealed a distinct signal co-localizing with the microvillous (apical) membrane of syncytiotrophoblasts. An analogous staining procedure using rabbit anti-GLUT9a antibodies also demonstrated the presence of hGLUT9a in microvillous membranes. Neither GLUT9a- nor GLUT9b was detectable in the basolateral membrane of syncytiotrophoblasts (Figure 1).

### 3.2. Transplacental Uric Acid Transport

To investigate the transplacental uric acid transport system, different models covering a wide range of maternal and fetal uric acid serum concentrations were used. In WT fetuses and in their mothers, uric acid levels were not different (n = 7, 27.43 ± 2.52 versus 28.25 ± 2.98 µmol/L, *p* = 0.70, NS, paired Student’s *t*-test). The serum levels in the fetal and maternal circulation ranging from 22 to 38 and 24 to 37 µmol/L, respectively, correlated very closely (n = 7, *p* > 0.001, r^2^ = 0.95, Figure 2A). To increase the severity of fetal hyperuricemia, maternal mice were challenged by a chow diet and inosine supplementation. Similarly, uric acid levels of WT fetuses and their mothers were not different (n = 7, 47.14 ± 6.60 versus 48.14 ± 4.85 µmol/L, *p* = 0.74, NS, paired Student’s *t*-test). The serum levels in the fetal and maternal circulation ranging from 29 to 65 and 32 to 63 µmol/L, respectively, correlated very closely (n = 7, *p* > 0.01, r^2^ = 0.85, Figure 2B). In contrast, under a normal chow diet, G9KO fetuses showed 4.95-fold higher uric acid serum levels than their mothers (n = 7, 142 ± 6.12 versus 29 ± 2.19 µmol/L, *p* < 0.001, paired Student’s *t*-test). Fetal serum levels correlated with those found in the maternal circulation (n = 7, ranging from 123 to 169 and 24 to 37 µmol/L, respectively, *p* < 0.005, NS, r^2^ = 0.86, Figure 3A). When challenged with inosine supplementation, WT fetuses showed 1.72-fold higher uric acid serum concentrations than fetuses from mothers under a normal chow diet (n = 7, 47.12 ± 6.60 versus 27.43 ± 2.52, *p* < 0.05, paired Student’s *t*-test). Serum levels in the fetal and maternal circulation ranging from 28 to 65 and 32 to 63 mmol/L, respectively, correlated closely (*p* < 0.01, r^2^ = 0.85, Figure 3B). Upon inosine supplementation, G9KO fetuses showed 4.10-fold higher uric acid serum levels than their mothers (n = 7, 192.88 ± 12.44 versus 47 ± 4.20 µmol/L, *p* < 0.005, paired Student’s *t*-test). Fetal serum levels did not correlate with those found in the maternal circulation (n = 7, ranging from 155 to 259 and 32 to 63 µmol/L, respectively, *p* = 0.12, NS, r^2^ = 0.35, Figure 3B).

## 4. Discussion

Hyperuricemia is a common laboratory finding in pregnant women compromised by preeclampsia. This circumstance is usually considered secondary to altered maternal kidney function. Hyperuricemia in preeclampsia, however, may also be explained by inflammatory and ischemic processes which subsequently increase uric acid production. A growing body of studies suggests that uric acid not only is a bystander but also plays an important role in the pathogenesis of preeclampsia [1,2]. Despite its important antioxidant function, uric acid has deleterious effects at higher concentrations or in a hypoxic environment. Several studies showed that elevated uric acid serum levels correlate with adverse maternal and perinatal outcomes [12,13,14,15]. The fetoplacental unit continuously produces uric acid which cannot be further metabolized in humans due to the mutational silencing of the liver enzyme uricase. To prevent its accumulation and its deleterious sequelae, uric acid has to be transported across the placenta into the maternal circulation, which allows renal secretion. In the proximal tubule of the kidney, GLUT9 transports uric acid across the basolateral membrane into the blood, as part of the reabsorption process, and thus it plays an important role in the homeostasis of serum uric acid levels in humans.

Knowledge regarding the placental uric acid transport system is scarce. In this study, we aimed to localize GLUT9 at different sites in the placenta. Immunohistochemical analysis and Western blotting initially failed due to the poor quality of commercially available antibodies against GLUT9. Thus, we raised specific antibodies against GLUT9a and GLUT9b in rabbits using N-terminal peptides of the GLUT9a and GLUT9b proteins. Immunohistochemical staining against placental GLUT9a and GLUT9b revealed distinct signals co-localizing with the microvillus (apical) membrane of syncytiotrophoblasts. Neither GLUT9a nor GLUT9b was detectable in the basal membrane of syncytiotrophoblasts. The simultaneous presence of GLUT9a and GLUT9b in the microvillus (apical) membrane of syncytiotrophoblasts may facilitate uric acid transportation from the fetoplacental unit into the maternal circulation, which is crucial for fetal well-being and survival. Our group has previously demonstrated that, in contrast to GLUT9b, GLUT9a is regulated by iodine [21]. The question of whether the localization of placental GLUT9a, i.e., on the microvillus membrane, is important for the regulation of transplacental uric acid transport capacity remains to be solved. Interestingly, we found both GLUT9 isoforms to be exclusively expressed in the microvillus (apical) membrane facing the maternal side, but not in basolateral membranes, of syncytiotrophoblasts. This is in contrast to the localization patterns of other cell types such as renal tubular cells, where GLUT9a and -9b are expressed in opposite plasma membrane domains. In tubular kidney cells, GLUT9a is expressed on the basal side, whereas GLUT9b was found to be expressed exclusively on apical membranes [22]. The distinct expression pattern in renal tubular cells ensures that GLUT9b reabsorbs uric acid from the urine, and that GLUT9a transports it across the basal membrane back into the circulation, thus maintaining uric acid homeostasis. Both GLUT9a and GLUT9b isoforms are high-capacity uric acid transporters. They are differentially expressed along the nephron but share similar transport capacities [16]. Since the transport characteristics of GLUT9a- and GLUT9b-mediated uric acid transport are congruent, the presence or absence of these isoforms in subcellular localization has an impact on transport capacity. Further, the presence of GLUT9a endorses the potential to regulate uric acid transport via iodine. Whether these regulatory mechanisms play a role in renal tubular cell re-uptake or in transplacental uric acid transport remains to be elucidated. The syncytiotrophoblast shows a specific expression pattern which, in turn, enables a unique transport pathway. The organic anion transporter-4 (OAT4) is another important placental uric acid transporter, which was shown to be expressed in the basal membranes of the syncytiotrophoblast. However, unlike GLUT9, OAT4 is unidirectional and only transports uric acid into the cell [23]. Our results suggest that uric acid is transported exclusively in the feto–maternal (but not in the materno–fetal) direction, which may help to protect the fetus from deleterious effects due to uric acid accumulation in the placenta.

To evaluate the placental potential to warrant uric acid homeostasis, we analyzed maternal and fetal uric acid serum levels in mice. We demonstrated that uric acid serum levels in the fetal and in the maternal circulation correlated very closely, even when uric acid levels were raised to supra-physiological levels by a diet with inosine supplementation. This observation highlights the ability of the placenta to efficiently maintain uric acid homeostasis between the maternal and fetal compartments using its uric acid transport system, in order to transport uric acid from the fetoplacental unit into the maternal circulation where it can be finally excreted by the kidneys and the intestines.

To assess the role of GLUT9 in the placental uric acid transport system, we used a murine G9KO model. In wild-type fetal mice, uric acid is transported by GLUT9 (which is expressed on hepatic cells) into these cells and, in turn, is further metabolized into allantoin by the liver enzyme uricase. Another pathway to prevent uric acid accumulation in these animals is transplacental uric acid transportation via GLUT9. Homozygous G9KO fetuses also lacking liver GLUT9, which prevents access to hepatic uricase, will develop hyperuricemia. Interestingly, these animals show exceedingly high uric acid serum levels. This finding indicates that the lack of placental GLUT9 is responsible for the inability to maintain placental uric acid homeostasis, and that GLUT9 is—at least in our animal model—the only relevant placental uric acid transporter. Our study has some limitations. First, due to the relatively small number of animals investigated, similar experiments with a larger sample size would corroborate our results. Second, data obtained from animal experiments cannot necessarily be extrapolated to human diseases.

Pro-inflammatory stimuli such as elevated uric acid levels during early pregnancy may facilitate the development of preeclampsia [24]. Further studies investigating the role of the placental uric acid transport system in preeclampsia are eagerly needed. These novel insights into the placental and renal uric acid transport systems may enable the development of therapeutic and preventive strategies for hyperuricemia-related diseases such as preeclampsia.

## Figures and Tables

**Figure 1 cells-11-00633-f001:**
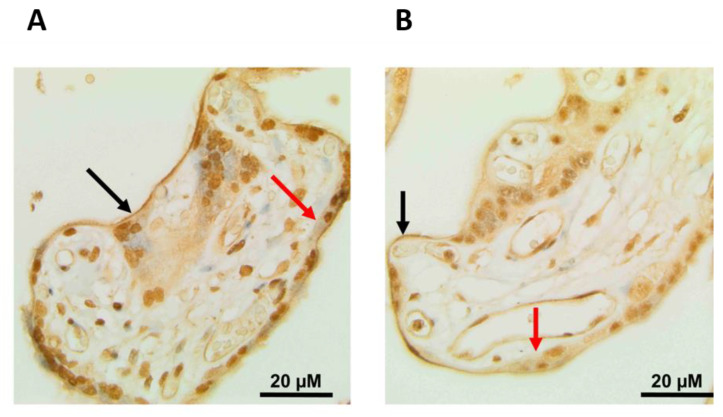
Immunohistological stainings of placental GLUT9a and GLUT9b. The black arrow marks the microvillus (apical) membrane, and the red arrow depicts the basal membrane of the syncytiotrophoblast in placental villi. Scale bar represents 20 µm. (**A**) Staining of a term placenta using anti-GLUT9a antibodies. (**B**) Staining of a term placenta using anti-GLUT9b antibodies. Both antibodies co-localize with the villous (apical) membrane of the syncytiotrophoblast, while no immunohistological signal is detectable in the basal membrane of the syncytiotrophoblast.

**Figure 2 cells-11-00633-f002:**
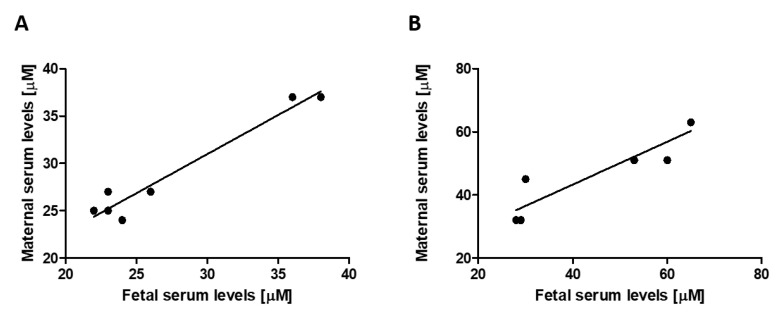
Correlation between maternal and (wild-type) fetal uric acid levels: (**A**) normal diet, and (**B**) diet with inosine supplementation.

**Figure 3 cells-11-00633-f003:**
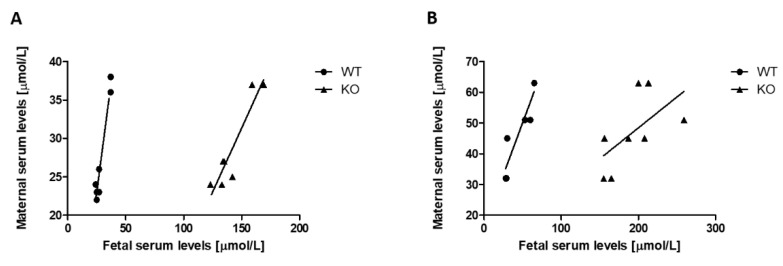
Correlation between maternal and (G9KO) fetal uric acid levels: (**A**) normal diet, and (**B**) diet with inosine supplementation.

## Data Availability

All data can be provided by the authors on request.

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
