# Peer review of "Glucose Transporter 9 (GLUT9) Plays an Important Role in the Placental Uric Acid Transport System"

_cells, 2022, doi:10.3390/cells11040633_

Round 1

Reviewer 1 Report

The authors Luescher et.al. in their manusript titled “ Glucose transporter 9 (GLUT9) plays an important role in the placental uric acid transport system”  investigated the role of Glucose transporter 9 in placental uric acid transport system. The authors have used placental tissue from human subjects as well as mouse model of Glut9KO to validate their finding that GLUT9 is responsible for increased uric acid levels in the condition of preeclampsia. The study is interesting, and important too and the authors have written manuscript in a succinct manner. It is easy to understand and follow the results.

There are few things that need to be addressed to make it clearer to improve the readability of the manuscript.

  1. Authors should mention more about different variants of GLUT9a and GLUT9b. Does these two isoforms perform two different functions? Or work together? Independently? How these two isoforms affect Uric acid levels?
  2. Please discuss more about the Glut9KO mice. What is the precise role and mechanism of uric acid levels in these mice.
  3. Are both the isoforms knocked out in Glut9KO mouse model?
  4. Is there any species level differences in GLUT9a and GLUT9b expression in mouse and human placental tissue?

Author Response

We would like to thank the reviewer for the important input. We would like to answer as follows:

The authors Luescher et.al. in their manusript titled “ Glucose transporter 9 (GLUT9) plays an important role in the placental uric acid transport system”  investigated the role of Glucose transporter 9 in placental uric acid transport system. The authors have used placental tissue from human subjects as well as mouse model of Glut9KO to validate their finding that GLUT9 is responsible for increased uric acid levels in the condition of preeclampsia. The study is interesting, and important too and the authors have written manuscript in a succinct manner. It is easy to understand and follow the results.

There are few things that need to be addressed to make it clearer to improve the readability of the manuscript.

  1. Authors should mention more about different variants of GLUT9a and GLUT9b. Does these two isoforms perform two different functions? Or work together? Independently? How these two isoforms affect Uric acid levels?

We added the following pragaraph tot he discussion:

“In tubular kidney cells GLUT9a is expressed on the basal side, whereas GLUT9b was found to be expressed exclusively on apical membranes [19]. The distinct expression pattern in renal tubular cells ensures that GLUT9b reabsorbs uric acid from the urine and that GLUT9a transports it across the basal membrane back into the circulation, thus, maintaining uric acid homeostasis. Both GLUT9a und GLUT9b isoforms are high-capacity uric acid transporter. They are differentially expressed along the nephron but share similar transport capacities [15]. Since the transport characteristics of GLUT9a- and GLUT9b-mediated uric acid transport are congruent, the presence or ab-sence of this isoforms on the subcellular localization has an impact on transport capacity. Further the presence of GLUT9a endorse the potential to regulate uric acid transport by iodine. Whether these regulatory mechanisms play a role in renal tubular cell re-uptake or in transplacental uric acid transport remains to be elucidated.”

  1. Please discuss more about the Glut9KO mice. What is the precise role and mechanism of uric acid levels in these mice.

To clarify the mechanism we added the following sentences in the discussion:

“In wild type fetal mice uric acid is transported by GLUT9 which is expressed on hepat-ic cells into these cells and in turn further metabolized into allontoin by the liver en-zyme uricase. Another pathway to prevent uric acid accumulation in these animals is transplacental uric acid transport via GLUT9.”

  1. Are both the isoforms knocked out in Glut9KO mouse model?

Yes, since we knocked out the GLUT9 gene, none of the splice variants is expressed in the homozygous G9KO fetuses.

  1. Is there any species level differences in GLUT9a and GLUT9b expression in mouse and human placental tissue?

To the best knowledge of the authors this is not known yet.

Reviewer 2 Report

This study investigated the location of two glucose transporter 9 (GLUT9) isoforms in human term placenta and investigated uric acid serum concentrations in pregnant GLUT9 knockout mice. While this study is of interest, it is written around the pregnancy disorder preeclampsia but none of the work is directly relevant to this disorder. Also, there is a great deal of relevant information missing. Please see below for more details.

Introduction:

The introduction is brief but does contain the required information to back up the study.

  • I recommend adding references to line 35.
  • Line 43 doesn’t make sense. Please rephrase.
  • How is this work related to preeclampsia? The introduction focuses on preeclampsia, yet none of the tissues, cells or animals have anything to do with this disorder.

Materials and methods:

There is a large amount of information missing from the materials and methods section.

  • Briefly detail how the HEK-293 cells were cultured.
  • Please detail how the placental tissue was handled i.e. how was it cut?
  • Add in n numbers throughout where relevant.
  • How were the negative controls performed for both 2.3 and 2.4?
  • Again, there is not enough information provided on the animals. Briefly detail how the mice were housed, the n numbers, a mating protocol etc. Did the treatment affect the fetuses i.e. was fetal size, number or number of reabsorbed pups different between treatments? Did placental weights differ? Was there no investigation as to whether these mothers were preeclamptic?
  • What is the rationale for that amount of inosine?
  • How was the blood processed? How was the uric acid measured? Plasma or sera? Was it diluted?

Results:

  • What is the rationale behind using the embryonic kidney cells? If it is simply to test the antibodies are working then it should be included in supplementary material.
  • Figure 2 please add in the negative controls.

Discussion:

  • What are the limitations of this study?
  • How are any of your results related to preeclampsia?

Author Response

We would like to thank the reviewer for the important input, which helped to improve the manuscript. We would like to answer as follows:

This study investigated the location of two glucose transporter 9 (GLUT9) isoforms in human term placenta and investigated uric acid serum concentrations in pregnant GLUT9 knockout mice. While this study is of interest, it is written around the pregnancy disorder preeclampsia but none of the work is directly relevant to this disorder. Also, there is a great deal of relevant information missing. Please see below for more details.

Introduction:

The introduction is brief but does contain the required information to back up the study.

  • I recommend adding references to line 35.

We have added the recommended references.

  • Line 43 doesn’t make sense. Please rephrase.

We have replaced the sentence:

“Plasma urate levels decrease in the first trimester of pregnancy by at least 25%, re-turning to normal level in the second trimester increases toward the end of pregnancy”

by

“Plasma urate levels decrease in the first trimester of pregnancy by at least 25%, return to normal level in the second trimester and increase toward the end of pregnancy.”

  • How is this work related to preeclampsia? The introduction focuses on preeclampsia, yet none of the tissues, cells or animals have anything to do with this disorder.

Elevated uric levels during early-pregnancy herald preeclampsia. Uric acid plays an important role in the pathogenesis of preeclampsia. Moreover children following pregnancies complicated by preeclampsia show impaired neonatal development. Novel insights to the transplacental uric acid transport system will enable potential preventive strategies for hypertensive pregnancy disorders.

Materials and methods:

There is a large amount of information missing from the materials and methods section.

  • Briefly detail how the HEK-293 cells were cultured.

We added:

“Human embryonic kidney cells (HEK-293) were cultered in minimal essential medium (Invitrogen) containing 10% heat inactivated fetal bovine serum albumin (FBS), 50 units/ml penicillin, and 50 μl/ml streptomycin. The cells were plated on 30mm dishes and transfected with a total amount of 1 μg of DNA/dish 3 μl lipofec-tamin 2000/dish (Invitrogen).

  • Please detail how the placental tissue was handled i.e. how was it cut?

We added the following sentence:

“The placenta was cut and washed in cold PBS. Villous tissue was dissected into 5-10 mm fragments for histological examination.”

  • Add in n numbers throughout where relevant.

We added the n numbers.

  • How were the negative controls performed for both 2.3 and 2.4?

We added the sentence:

“Negative control performed in absence of primary antibody revealed no detectable signal.”

  • Again, there is not enough information provided on the animals. Briefly detail how the mice were housed, the n numbers, a mating protocol etc. Did the treatment affect the fetuses i.e. was fetal size, number or number of reabsorbed pups different between treatments? Did placental weights differ? Was there no investigation as to whether these mothers were preeclamptic?

The treatment did not affect litter size. The mating of heterozygous GLUT9+/- animals results in a typical Mendelian off-springs ratio of 25:50:25 for (wild type):(hetero-zygous):(knock out). To weigh the placentae was technically impossible because they were too small. The mothers were heterozygous and therefore normal uric acid levels. However we have not checked for hypertension or proteinuria in these animals.

We added the paragraph:

“Mice where kept at 12hr day/night cycles in colonies of 5 mice per cage with free access to food and water. Female mice where prepared for mating by placing the animals in a cage where male animals where kept before, 3 days ahead of mating. For the mating 2 females animals where placed together with 1 male animal. On the next day at 7:30 AM the animals were separated and females where checked for plaque formation. Detection of a plaque was defined as day 0.5 of carriage. The mating of heterozygous GLUT9+/- animals results in a typical Mendelian off-springs ratio of 25:50:25 for (wild type):(hetero-zygous):(knock out). ”

  • What is the rationale for that amount of inosine?

We wanted to induce elevated uric acid levels by an altered diet. The used amount of inosine raised sufficiently the uric acid levels. The other possibility would have been by intraperitoneally injected inosine.

  • How was the blood processed? How was the uric acid measured? Plasma or sera? Was it diluted?

We added the following sentence:

“Plasma uric acid was analyzed using the Roche/Hitachi 902 robot system (Roche); no dilution of fetal sera was needed.”

Results:

  • What is the rationale behind using the embryonic kidney cells? If it is simply to test the antibodies are working then it should be included in supplementary material.

We included this section in supplementary material as suggested by the reviewer.

  • Figure 2 please add in the negative controls.

We added the negative control in supplementary material.

Discussion:

  • What are the limitations of this study?

We added:

“Our study has some limitations. First due to the relatively small number of animals investigated similar experiments with a larger samples size would corroborate our re-sults. Second, data obtained following animal experiments can not necessarily be ex-trapolated to human diseases.

  • How are any of your results related to preeclampsia?

It widely accepted that any increased inflammatory state during early-pregnancy will facilitate the development of preeclampsia. Therefore the understanding of the exact meachanisms of placental and renal uric acid transport system is a prerequisite for the development of novel therapeutic and preventive strategies to cure uric acid-associated diseases.

We reformulated in the discussion as follows:

“Pro-inflammatory stimuli such as elevated uric acid levels during early-pregnancy may facilitate the development of preeclampsia. Further studies investigating the role of the placental uric acid transport system in preeclampsia are eagerly needed. The novel insights into the placental and renal uric acid transport sys-tem may enable the development of therapeutic and preventive strategies for hyperu-ricemia-related diseases such as preeclampsia.”

Round 2

Reviewer 2 Report

I am happy with the responses. Thank you for taking the time to revise the manuscript accordingly.